# Delivering genome sequencing in clinical practice: an interview study with healthcare professionals involved in the 100 000 Genomes Project

Saskia C Sanderson,[1,2] Melissa Hill  ,[1,3] Christine Patch,[4,5,6] Beverly Searle,[7] Celine Lewis,[1,3] Lyn S Chitty[1,3]

SCS and MH are joint first authors.
CL and LSC are joint senior authors.

For numbered affiliations see end of article.

**Correspondence to**
Dr Melissa Hill;
melissa.hill@ucl.ac.uk

## ABSTRACT

**Objectives** Genome sequencing is poised to be incorporated into clinical care for diagnoses of rare diseases and some cancers in many parts of the world. Healthcare professionals are key stakeholders in the clinical delivery of genome sequencing-based services. Our aim was to explore views of healthcare professionals with experience of offering genome sequencing via the 100 000 Genomes Project.

**Design** Interview study using thematic analysis.

**Setting** Four National Health Service hospitals in London.

**Participants** Twenty-three healthcare professionals (five genetic clinicians and eight non-genetic clinicians (all consultants), and 10 'consenters' from a range of backgrounds) involved in identifying or consenting patients for the 100 000 Genomes Project.

**Results** Most participants expressed positive attitudes towards genome sequencing in terms of improved ability to diagnose rare diseases, but many also expressed concerns, with some believing its superiority over exome sequencing had not yet been demonstrated, or worrying that non-genetic clinicians are inadequately prepared to discuss genome sequencing results with patients. Several emphasised additional evidence about utility of genome sequencing in terms of both main and secondary findings is needed. Most felt non-genetic clinicians could support patients during consent, as long as they have appropriate training and support from genetic teams. Many stated genetics experts will play a vital role in training and supporting non-genetic clinicians in variant interpretation and results delivery, particularly for more complex cases.

**Conclusions** Healthcare professionals responsible for delivering clinical genome sequencing have largely positive views about the potential for genome sequencing to improve diagnostic yield, but also significant concerns about practical aspects of offering these tests. Non-genetic clinicians delivering genome sequencing require guidance and support. Additional empirical evidence is needed to inform policy and practice, including how genome compares to exome sequencing; utility of secondary findings; training, in particular of non-genetic health professionals; and mechanisms whereby genetics teams can offer appropriate support to their non-genetics colleagues.

## Strengths and limitations of this study

► The qualitative methodology employed in this study facilitated an in-depth exploration of interviewees' perspectives and allowed them to express their views in their own words; however, the sample size was small and encompassed only four London hospitals which brought limitations for wider generalisability and did not allow between group comparisons.

► Some of the interviewees were 'consenters' from the 100 000 Genomes Project so although they had real experience of supporting patients during the informed consent process, most would not be delivering genome sequencing clinically in the future.

► Interviewees also included a range of genetic and non-genetic healthcare professionals from multiple hospital sites who *will* be responsible for delivering genome sequencing in the new UK Genomic Medicine Service.

► Interviewees' views were informed by their experience of actually offering genome sequencing to National Health Service patients, rather than views being entirely based on hypothetical scenarios; however, at the time of the interviews very few results had been returned from the 100 000 Genomes Project.

► Our findings shed important light on where the evidence gaps about clinical genome sequencing are from healthcare professionals' perspectives; this information may also be of use to practitioners and policy-makers concerned with delivering genome sequencing clinically going forwards.

## BACKGROUND

Genome sequencing is increasingly being incorporated into clinical care for the purposes of diagnosing rare disease and some cancers. Genome sequencing allows us to determine most of a person's complete set of DNA by interrogating both the coding and non-coding regions of the genome. This contrasts with exome sequencing which only looks at the coding regions of the genome or

more targeted approaches that use gene panels to focus on specific sub-sets of candidate genes. Exome sequencing and panel testing have been widely used in both research and clinical settings, but as costs come down greater use of genome sequencing is occurring. Genome sequencing is more comprehensive than exome sequencing, with wider genome coverage and growing evidence of improved diagnostic yield in some contexts.[1] However, the potential for genomic sequencing to identify a wider range of variants as well as secondary findings that are unrelated to the indication for testing is a key consideration for implementation as this adds to the complexity of consent and return of results processes.

In the National Health Service (NHS) in England a new Genomic Medicine Service was launched in October 2018 that will include offering genome sequencing for patients with undiagnosed rare genetic diseases or a current diagnosis of cancer. A National Test Directory has been developed that describes the full range of tests available. The eligibility, testing and consent materials will be standardised across the Genomic Medicine Service.[2] The springboard for this new service had been the 100 000 Genomes Project which, under research consent and governance, recruited patients with rare diseases (children and adults) and their relatives, and patients with cancer, for genome sequencing between 2015 and 2018. Recruitment was conducted through 13 NHS Genomic Medicine Centres (GMCs) located across the UK. In addition to receiving the main findings relevant to the indication for testing, participants could also opt in to secondary findings being looked for by the project scientists. The list of looked for secondary findings includes variants in genes that increase predisposition to breast and ovarian cancer (including *BRCA1* and *BRCA2*), Lynch syndrome (colorectal cancer and other cancers) and familial hypercholesterolaemia. Participants could also opt in to secondary analysis of their genomic data to determine carrier status for cystic fibrosis. The return of main and secondary findings from the 100 000 Genomes Project is ongoing and participants are being given their results through NHS clinical pathways.

In England as elsewhere, there are efforts to mainstream genomic testing such that, when indicated, both genetic and non-genetic clinicians will be able to order genome sequencing for their patients, as they would other clinical tests. The main stated benefits of genome sequencing for rare disease diagnoses include increased discovery rate of causative gene variants, the potential to dramatically shorten the 'diagnostic odyssey', preventing multiple, often invasive and expensive, investigations and allowing targeted therapies.[3] However, moving genome sequencing from research to clinical practice raises many challenges. These include how to ensure there is the relevant expertise and infrastructure to deliver clinical genome sequencing nationally; whether non-genetic clinicians need support to offer genome sequencing and return results to patients as genomics is mainstreamed, and if so, what kind of support; and what results should

be returned and to whom? While many of these questions arise in the context of other types of genetic tests, particularly exome sequencing, the potential challenges are even greater in the context of genome sequencing because of the sheer volume of data generated for any given individual or family.

Healthcare professionals are key stakeholders in answering these questions given they are central to offering genome sequencing to patients, facilitating informed consent, supporting patients' decision-making and returning results to patients. Previous studies have explored healthcare professionals' experiences of obtaining informed consent for genomic sequencing[4–6] and returning genomic results,[7 8] and views regarding secondary findings from genomic sequencing.[9 10] Most of these studies were conducted in the USA,[4–9] one in Canada[10] and the only UK study to date drew on experiences of a single multidisciplinary team (MDT) and focused exclusively on secondary findings.[11]

The 100 000 Genomes Project affords a unique opportunity to gather stakeholder views on the implementation of genome sequencing in the setting of a large scale, nationally implemented initiative where healthcare is state funded and the intention is to return clinically relevant diagnoses. Here we aim to explore views of a range of healthcare professionals involved in the 100 000 Genomes Project regarding moving genome sequencing for rare disease diagnoses into clinical practice within the new Genomic Medicine Service, with a view to providing insights that may be relevant to policy and practice.

## Methods
### Study design
A qualitative approach using one-to-one interviews for data collection was used to facilitate in-depth exploration of healthcare professionals' viewpoints on moving genome sequencing into NHS clinical practice.

### Patient and public involvement
The advisory team for this study had one lay-person and three patient support group representatives; including co-author BS, who has also contributed to the revision of the manuscript. Over the course of the study the advisory team has provided ongoing review and feedback on study conduct, research materials, interpretation of data and reporting of findings.

### Sampling and recruitment
Participants were recruited from four London hospitals that were part of two Genomic Medicine Centres (GMCs) recruiting affected individuals and their relatives into the 100 000 Genomes Project. Three groups of health professionals were recruited to the study: (1) genetic and (2) non-genetic clinicians involved in identifying patients eligible to take part in the 100 000 Genomes Project; and (3) 'consenters', that is, members of the 100 000 Genomes Project consenting teams whose role included consenting patients for genome sequencing. Potential participants in

this study were identified by our research team and invited to participate via an email which included participant information explaining the aims of the study (see online supplementary material). They were asked to contact the researchers (CL, MH, SS) if interested in participating. Written consent was obtained prior to the interviews to ensure informed and voluntary participation.

Participants were purposefully sampled across four London hospitals to ensure inclusion of participants from different geographical locations, various approaches to recruitment and clinicians from a range of clinical backgrounds. Consenters in the 100 000 Genomes Project have a range of backgrounds, including genetic counselling, research and other post graduate training. All consenters are trained, including taking the online course, 'Preparing for the consent conversation' (https://www.genomicseducation.hee.nhs.uk/courses/courses/consent-ethics/). Interviews and analysis were undertaken concurrently, and recruitment ceased when no new themes were emerging during the interviews.

### Interviews

An interview guide was developed for this study by the investigators to explore participants' experiences and views on (a) recruitment into and (b) return of results in the 100 000 Genomes Project, as well as their views on moving genome sequencing into clinical practice, and their professional background and training for the 100 000 Genomes Project. To develop the interview guide, a first draft of a set of questions reflecting the aims of the project was drafted by one of the investigators (MH) with experience in qualitative research methods; this was iteratively refined via discussion with two other investigators (CL, SS) on the team also with experience in qualitative methods; minor revisions were then made to the interview guide in response to questions that emerged during the first two interviews. There were two versions of the interview guide: one for clinicians and one for consenters (see online supplementary material). The consenter version included additional sections about views on more specific aspects of informed consent and patients' experiences. The interviews were conducted face-to-face or via telephone by CL, MH or SS.

### Data analysis

Interviews were digitally recorded, transcribed verbatim and anonymised. Pseudonyms were then assigned to each participant. Data were analysed using the principles of thematic analysis.[12] NVivo V.10 (QSR International, Australia) software was used to facilitate coding and data analysis.

The focus of the analyses was text segments relevant to moving genome sequencing for rare disease diagnosis into clinical practice, reflecting the aims of this study. The research team comprised three postdoctoral researchers (SCS, CL, MH) with expertise in conducting qualitative analyses, two genetic healthcare providers (CP, LSC) with experience supervising qualitative analyses and one

patient advocate (BS). The transcripts were coded by SCS and MH. The two investigators independently coded the same transcript and developed a draft codebook, and the two versions of the codebooks were compared and combined into a single codebook. The two investigators then independently coded a second transcript: a Kappa was calculated and indicated good inter-rater reliability. From this point the two investigators independently coded the remaining transcripts into eight broad overarching categories. The overarching category specifically focused on moving genome sequencing into routine clinical practice was then coded line by line into meaningful units of text. To ensure rigour and increase authenticity, multiple investigators (SCS, CL, MH) with varying levels of familiarity with genome sequencing checked the emerging themes at multiple time points and together grouped the themes to form broader themes that were reviewed and redefined as the analysis progressed.

## RESULTS
### Participants

The 23 participants were 5 genetic clinicians (all consultants in clinical genetics), 8 non-genetic clinicians (all consultants, from a range of specialities including audiology, paediatric nephrology, neurology and paediatrics) and 10 consenters (with a range of backgrounds, including nursing, project management, postdoctoral research and medical doctor trainees) from four hospital sites. See table 1 for additional participant characteristics. There were seven telephone and 16 face-to-face interviews. The interviews ranged in length from 19 to 75 min (median=34 min).

### Overarching themes

Participants' views on moving genome sequencing for rare disease diagnosis into clinical practice fell into three broad overarching themes; 1. Attitudes towards moving genome sequencing into clinical practice, 2. Attitudes towards offering secondary findings from genome sequencing and 3. Views on how genome sequencing should be delivered in clinical practice.

### Attitudes towards moving genome sequencing into clinical practice
#### Perceived benefits

Participants talked about three main ways in which clinical genome sequencing will benefit patients. First, some participants felt that clinical genome sequencing will lead to improved diagnoses for patients with rare diseases by making the path to diagnosis more efficient, making it more likely patients will get a diagnosis with fewer tests needed. Second, some participants felt that clinical genome sequencing will add to the knowledge base about rare diseases. Third, several participants talked about how research will lead to improved treatments for future patients, either by improving disease progression stratification or by personalising treatments based on

**Table 1** Participant characteristics

| Characteristic | N (%) |
|---|---|
| Gender | |
| Female | 19 (82.6) |
| Male | 4 (17.4) |
| Age | |
| 20–30 years | 7 (30.4) |
| 31–40 years | 1 (4.3) |
| 41–50 years | 8 (34.8) |
| 51–60 years | 7 (30.4) |
| Role | |
| Genetic clinician | 5 (21.7) |
| Non-genetic clinician | 8 (34.8) |
| Audiovestibular medicine | 2 |
| Neurology | 2 |
| Neuromuscular | 2 |
| Nephrology | 2 |
| Consenter | 10 (43.5) |
| Nursing | 3 |
| Project management | 1 |
| Laboratory research scientists | 2 |
| Medical doctor trainees | 1 |
| Other | 3 |
| Completed formal online training for 100 000 Genomes Project | |
| Yes | 13 (56.5) |
| No | 10 (43.5) |

medication responsiveness (pharmacogenomics). See table 2 for illustrative quotes.

## Concerns

Although many participants expressed positive attitudes towards genome sequencing, some also raised concerns. First, a few participants expressed concern that genome sequencing has not been demonstrated to be superior to exome sequencing for diagnosing rare diseases, and believed genome sequencing should not be introduced clinically until there is scientific evidence about its value. Second, many participants had concerns about informed consent, particularly around the time needed for this. Participants emphasised that the consent form for the 100 000 Genomes Project was very long, and moreover that consent discussions had been undertaken by a dedicated consent team who would no longer be available after the transition to clinical practice. Some were concerned that some non-genetic clinicians do not understand the limitations of genome sequencing sufficiently, and that they are therefore under-prepared to offer genome sequencing to their patients, for example:

"…having overheard conversations from colleagues, I don't think they fully understand the limitations on what they are offering. I think clarity as to what this test will tell you, what it will not tell you. I think that needs to be clarified. And I think the colleagues need to understand before they offer it to their patients." (Participant 12, genetic clinician)

Third, participants voiced concerns about whether adequate resources—including laboratory and clinician time and expertise—will be available to analyse and interpret sequence data and subsequent variant validation, with several feeling genetic departments will be 'swamped'. Participants also raised concerns about phenotyping taking up the time of non-genetic clinicians, for example:

"presumably we would still have to do the phenotype data and certainly for clinicians, that's the limiting step, filling out another form!"(Participant 23, non-genetic clinician)

Concerns were also expressed about the potentially very long turn-around times to results. Several participants were also concerned that reports generated by labs were not 'filtered' enough, meaning a lot of work was still left for clinicians to do. Fourth, a few participants expressed concerns about disclosing results to patients both in terms of the amount of consultation time needed to discuss results with patients and families, and whether non-genetic clinicians have sufficient understanding to be able to interpret and return more complex results. See table 2 for additional illustrative quotes.

### Attitudes towards offering secondary findings from genome sequencing
#### Perceived benefits
Participants perceived three main benefits of offering secondary findings from genome sequencing in clinical practice. First, several participants felt that offering clinically actionable secondary findings in the clinical setting will provide the opportunity to identify patients at increased disease risk, allowing for improved disease prediction, prevention or early diagnosis. Second, several participants felt that including secondary findings will lead to research learning opportunities: for example, how we will learn more about the medical, social and financial impact of secondary findings on patients. Third, several participants felt that patients want or would want to receive secondary findings, suggesting that offering this type of information might be responsive to patients' desires and preferences. See table 3 for illustrative quotes.

#### Concerns
The issue of whether secondary findings should be offered to patients divided participants, with some advocating a cautious adoption while collecting data on penetrance and outcomes, and others stating that secondary findings should not be offered at all. First, some were concerned that evidence is lacking regarding the penetrance of

**Table 2** Health professionals' attitudes (perceived benefits and concerns) towards moving genome sequencing into clinical practice

| Content topic | Illustrative quote |
|---|---|
| **Perceived benefits** | |
| 1. Improved diagnoses for patients | "obviously we will get lots more diagnoses and that's good, I think that's good for everyone… and I think it helps, it helps for counselling, prevention, this type of things" (Participant 16, non-genetic clinician) |
| 2. Contributions to knowledge base | "It's a technology that gives us access to, you know, vast quantities of information and if we as clinicians and scientists are able to interpret that in a meaningful way… and the more information we gather, then the easier it should be to interpret because we've got so many comparisons and so much more data to base our decisions on." (Participant 18, non-genetic clinician) |
| 3. Improved treatments for future patients | "…so if we can really correlate the clinical information with the genetic information and find these markers that will allow this personalised treatment, then we have a big step forward…." (Participant 11, non-genetic clinician) |
| **Concerns** | |
| 1. Lack of evidence | "I think it's too early to know whether or not (whole genome sequencing) should become a routine part of clinical practice. I think we've now got sufficient evidence that whole exome sequencing can be very useful clinically… my own view is that we shouldn't roll out whole genome sequencing clinically until we have objective scientific evidence that it's superior to whole exome sequencing." (Participant 14, genetic clinician) |
| 2. Informed consent | "What concerns me is that every single member of the 100 000 genomes team has said to me that it's an hour to do the full consent process… I think most clinicians don't have an hour spare to be going through that with patients… I think it's very difficult for clinicians to do that genetic counselling. I don't think it will be done particularly well, because it certainly won't, it will be a 5 min process." (Participant 17, non-genetic clinician) |
| 3. Resources for analysis and interpretation | "It takes a lot of lab time to look at the data, it takes a lot of clinician time, to prepare the cases for a multidisciplinary team meeting…" (Participant 13, genetic clinician) |
| 4. Interpretation and disclosure of results | "I think the problem is that other doctors think they're trained but I don't think they are… I'm much less confident about explanations of results… one sees all the time problems that arise because of that so results are over interpreted… so I am a bit worried about that kind of thing. Because I already see it…" (Participant 13, genetic clinician) |

variants identified 'completely incidentally' and that we may be 'over-estimating penetrance' as a consequence.

Related to this, the second sub-theme that emerged was a concern that evidence is lacking regarding whether it is 'clinically useful' to return secondary findings to patients, particularly if they do not have a known family history of the associated condition. One participant said that this 'opens a can of worms'.

The third subtheme that emerged was concern about the potential psychological impact of the results on participants, such as causing 'anxiety'. One participant asked, "How do you stop people from suffering unduly from getting these results back?" (Participant 8, consenter), while another asked, "are you opening up Pandora's box?" (Participant 5, consenter). One participant likened secondary findings to picking up aneurysms incidentally on brain scans (see table 3).

The fourth sub-theme that emerged was a specific concern that non-genetic clinicians are not prepared to discuss secondary findings. One participant was concerned that doctors without 'a genetics training background' might misinterpret secondary findings and that this could lead to 'false reassurance' (Participant 6, consenter). Another emphasised that non-genetic clinicians couldn't decide whether mutations were 'real' or what the associated disease 'risk' would be (see table 3).

Related to this concern, some participants felt that, if secondary findings were to be offered, return of results should only be done by genetic experts and/or via Genetics Departments. One participant stated that he felt it would be 'completely unsafe' for non-genetic clinicians to offer such findings to patients. Some also felt that Genetics Departments already have referral pathways in place (eg, cancer screening) and that this was another reason that secondary findings should only be offered by genetic clinicians, if at all.

Fifth, some participants were concerned about what their clinical recommendations for their patients would be, and how they would advise their patients, based on secondary findings. Linked to this, some participants were concerned about whether the 'list' of secondary findings would change and whether patients would be 're-tested' if/when that did happen.

The final sub-theme that emerged, reflecting all of the concerns above, was that it would be more straightforward if secondary findings were not offered in clinical practice. Several participants stated that clinicians already offer clinical exome sequencing and that secondary findings are not offered as part of that clinical service in the UK. These participants felt that the same approach should be taken with clinical genome sequencing. See table 3 for illustrative quotes.

**Table 3** Health professionals' attitudes (perceived benefits and concerns) towards secondary findings

| Content topic | Illustrative quote |
|---|---|
| **Perceived benefits** | |
| 1. Improved prediction, prevention or early diagnosis of complex diseases for current patients | "…and also I think it's part of offering, it's part of helping people to get healthier. If we can prevent things happening or help them earlier in the stage of something happening, I think it's part of our duty to actually do that." (Participant 7, consenter) |
| 2. Opportunity to advance research | "And as I said as we learn more we're going to appreciate number one how people receive that information and the impact it has on them, sort of medically as well as socially, financially et cetera…. So yeah I think it's just going to be a moving field and we're going to learn from mistakes, but you know probably gain some of the understanding." (Participant 18, non-genetic clinician) |
| 3. Being responsive to what patients want | "Yeah I think you know, if there's something you can do about it, then I think that's fine yes. I think a lot of people would want to know and they can be consented up front." (Participant 22, genetic clinician) |
| **Concerns** | |
| 1. Evidence is lacking regarding the penetrance of variants identified incidentally | "…the principle I think is sound but I think what we may find is that the penetrance of a lot of these things is not as high as we thought it was and that maybe if, if you ascertain somebody as a carrier of a pathogenic mutation completely incidentally what does that mean in terms of penetrance, I think we may be over, over estimating penetrance…" (Participant 13, genetic clinician) |
| 2. Evidence is lacking regarding whether it is clinically useful to return secondary findings | "I think they (secondary findings) should be done when it's clinically necessary not just for the hell of it" (Participant 4, consenter) |
| 3. Potential psychological impact of the results | "…even when you do find your aneurism and… you tell them that their risk of developing something is very low but they tend to walk around thinking they've got a time bomb in their head anyway and the same [is] going to apply with genetics. Incidental findings as well probably much more so in fact." (Participant 19, non-genetic clinician) |
| 4. Non-genetic clinicians are not prepared to discuss secondary findings with patients | "…if we were to offer this for routine clinical practice, you know, in a neurological hospital how can we possibly decide whether, you know, a mutation in a cancer gene is a polymorphism or real or what's the risk you know? We cannot counsel patients on all these other things." (Participant 21, non-genetic clinician) |
| 5. Not clear what clinical recommendations for patients would be based on some secondary findings | "I think in BRCA and the MMR gene there's plenty of evidence out there. So you know you'd have to have [screening], so it needs to be recurrent mutations that are definitely associated and so what do you do then if you go and find something else. I don't know whether these people should be on screening or not " (Participant 9, genetic clinician) |
| 6. More straightforward if secondary findings were not offered as part of genome sequencing in clinical practice | "Yeah well again I would not treat this as any way different than to the clinical exome we do here or the exome we do on research. You look at the things that you are interested in." (Participant 11, non-genetic clinician) |

### Views on how genome sequencing should be delivered in clinical practice

Participants' views on genome sequencing in clinical practice fell into three overarching themes.

#### Non-genetic clinicians can offer genome sequencing in clinical practice *as long as they have appropriate training and are supported by healthcare professionals with genetics expertise*

Participants felt that non-genetic clinicians should be able to offer genome sequencing to patients in their clinical practice but that adequate training and support from genetics teams were critical (table 4). One participant stated that ideally consent would be conducted by a genetic counsellor but that this is not practical. For many participants, appropriate training was key although most felt that the consent process did not need to be conducted by a genetic expert, they also felt that anyone offering genome sequencing consent to patients should only do so after adequate training.

One participant emphasised that non-genetic clinicians can take responsibility for the consent process for genome sequencing 'but there needs to be the appropriate support systems in place', and another suggested that clinicians could consent patients but that patients might at least need the 'option' of being able to talk to and ask questions of a genetic counsellor, genetic nurse or other genetic specialist at the time of consent.

Many participants felt that non-genetic clinicians should be involved in the return of results to patients, but highlighted that they would need support from genetics departments/teams in interpreting and discussing complex findings, including secondary findings and more complex main findings. Notably, in this context, several clinicians spoke of existing working relationships between genetics and other specialty groups and the importance of working in MDTs, for example:

"they know we'll be around when the results come back if they're not clear cut and they need somebody

**Table 4** Health professionals' views on whether/how genome sequencing should be delivered by non-genetic clinicians in clinical practice

| Content topic | Illustrative quote |
|---|---|
| Non-genetic clinicians can offer genome sequencing in clinical practice *as long as they have appropriate training and are supported by healthcare professionals with genetics expertise* | |
| 1. Not practical to have genetics experts (eg, genetic counsellors) to offer & return all results | "I mean in an ideal world actually what you'd do, what you would have is someone additional in your clinic perhaps a Genetic Counsellor who was working in the clinic who could do that kind of thing, but that's not how it works, it's not how we deploy our resources… I think it will be offered through other health professionals…" (Participant 13, genetic clinician) |
| 2. Non-genetic clinicians can offer genome sequencing as long as they have adequate training | "With appropriate training I think it probably could be offered by, by appropriate professionals…if it was a hospital Consultant and they had the time to really discuss it and they had the appropriate knowledge and training and the confidence to give the results back then I don't see that being a problem…" (Participant 6, consenter) |
| 3. Non-genetic clinicians can offer genome sequencing as long as they have sufficient support from genetic departments / teams in interpreting, returning and/or discussing complex findings | "I think (offering genome sequencing) does need to be linked closely with genetics… I think that's key because otherwise you could potentially think something is relevant and pathogenic when actually it might not be or it's complicated with another variant… we must never underestimate the complexity of it… So I suppose that means that you would and should need access to your genetics team." (Participant 23, non-genetic clinician) <br> "So if you don't know anything about the genes then obviously the discussion of the findings should be left to somebody who has experience of genetics I do think yes." (Participant 11, non-genetic clinician) |

to discuss it with because they do that all the time with other tests that they request… They know we're here to help with the interpretation of that, they just need to know enough to be able to offer the right genetic test to the right patient, at the right time." (Participant 14, genetic clinician)

### Views on training content

Participants made a number of recommendations about what training non-genetic clinicians will need in order to ensure they have adequate knowledge to receive informed consent for genome sequencing from patients. In addition to having a knowledge of basic genetics, they would need to emphasise that genome sequencing is optional, manage patients' expectations regarding turnaround time for results, and admit if they do not know the answer to a patient's questions. Participants also felt that non-genetic clinicians need guidance regarding who should be offered genome sequencing. The recommendations for training content are summarised in table 5.

### Views on training delivery

In addition to the content of the training, participants expressed views about how training for non-genetic clinicians might best be delivered, with several suggesting that all doctors and nurses should be trained at medical or nursing school as part of the curriculum. Participants also suggested that online training was helpful for some types of learning (eg, basic genetics) and that this should be interactive (eg, with videos) rather than only static written information, but that other types of learning (eg, counselling) needed to be in-person (eg, shadowing). Several people felt that non-genetic clinicians would be most likely to attend half-day or 1-day training days or

study days, rather than longer modules. Several participants also felt that such training should be mandatory (including regular and refresher training) and that there should be some kind of 'National Standard', 'certificate' or 'certification'. Finally, several participants talked about the key role that genetic experts are already playing and will continue to play in training and 'upskilling' their non-genetic colleagues, by working closely with them and supporting them, and via 'buddying systems' and 'genomics champions'. See table 6 for further details and additional illustrative quotes.

### DISCUSSION

In this study, we found that healthcare professionals with experience of delivering genome sequencing in the UK NHS as part of the 100 000 Genomes Project had mixed views regarding moving genome sequencing into clinical practice. Positive views were expressed regarding the potential for genome sequencing to improve diagnoses of rare diseases, contribute to the rare disease knowledge base and lead to improved treatments in the future. However, some concerns were also expressed. Some participants' worries were primarily about whether healthcare professionals in genetics and non-genetics departments have adequate resources to explore the tiered variant report that is returned, undertake variant interpretation, conduct additional phenotyping and technically validate results to issue a report. They also had reservations as to whether non-genetic clinicians are adequately prepared to disclose results to patients. Similar concerns about whether clinicians, particularly non-genetic clinicians, have the necessary resources and training to include genome sequencing in their practice have been reported

**Table 5** Health professionals' views/recommendations for <u>content</u> of training for non-genetic clinicians

| Content topic | Illustrative quote |
| --- | --- |
| Basic genetics | "All doctors need to be trained in the basics of genetics" (Participant 11, non-genetic clinician) |
| Guidance regarding who should be offered whole genome sequencing | "I think we should, we need to have guidelines as to who should be the right candidate and as long as we have, we know that, then I think any professional would be able with the requisite training and who's informed about the indications, but(…) then yes they should be able to offer it." (Participant 15, non-genetic clinician) |
| Limitations/process/managing patients expectations | "So I think it's essential that you understand the process, you understand the limitations… you have to give them honest expectations of process. And I think that's your own responsibility to try and understand the science of it. So yeah I think that's the most important thing not to give people false hope, not selling this as some magic new technology to give the answer to a problem. And the difficulty of interpretation and just why it takes so much time." (Participant 18, non-genetic clinician) |
| Build confidence in ability to answer patients' questions | "if you are not confident you cannot discuss genetic, because parents will ask questions. If you don't know the answer, then not going to work. You have to be confident when you have these discussions with parents I think." (Participant 13, genetic clinician) |
| Admit if don't know answers to patients' questions | "The offering it to them, if you're not sure of the answer tell them that you will find out, don't make it up." (Participant 21, non-genetic clinician) |
| Emphasise patient has the option to decline | "I think what's happening also sometimes, when a consultant offers something to the patient, whatever it is, the patients are very keen to say yes because it's their specialist telling them something and rightly they think it's an important thing, but they don't always know that they have the right or the option to actually refuse it. So just to stress to the consultant that they have to make that option very clear. This is a great test which I agree with but "You don't have to take part" usually is left out because we don't have time to do that. And I would love a little bit less of "Great" and a little bit more of "You have the option of not taking part"." (Participant 7, consenter) |
| Allow adequate time to inform patients properly and to make their decision | "what we offer them has to be explained properly so they can make an informed consent and not using that to our advantage, just to get someone to consent into a study… What we're offering, we think it's this and that, and that, make it very clear, and let them make the informed consent and to take the decision in their time. Some people might need a little bit more time than others…" (Participant 7, consenter) |
| How to understand/deal with a genetics report | "How do you annotate a variant? What does it mean if it's a class one, two, three, four, five, and how do you interpret that?(…)So I think these are things that just need to be clear to everybody how to understand a genetic report, what does make a class three, class four, whatever, but I think that's any clinician though." (Participant 11, non-genetic clinician) |

in a US study exploring health professional experiences of returning genome sequencing results.[8] Participants in the present study were also concerned about secondary findings from genome sequencing being offered in clinical practice. Although perceived benefits of offering secondary findings that included improved prediction and prevention of complex diseases, advancing research and being responsive to what patients want were discussed, a number of concerns were also voiced. Concerns about reporting secondary findings to patients included lack of evidence regarding the possibly lower penetrance of variants identified incidentally, lack of evidence regarding whether it is clinically useful to return such results, how clinicians should advise patients to act based on the results, the potential for such findings to cause patients anxiety and again that non-genetic clinicians do not have the necessary expertise to counsel patients about secondary findings. Overall, however, participants felt that non-genetic clinicians *will* be able to offer genome sequencing to patients in clinical practice, *as long as* they have adequate training and support from colleagues with expertise in genetics, particularly for more challenging cases and complex findings. They also offered specific recommendations for what that training and support should look like and how it should be delivered.

Our findings have implications for clinicians and policy-makers concerned with moving genome sequencing into clinical practice. Our findings suggest that non-genetic clinicians will be able to support patients in their decision-making, receive consent and order genomic sequencing tests as long as they have adequate training and support from genetic teams to help with interpretation of results and explanations to patients in complex cases. Clear protocols and processes for referral to genetics for patients to discuss complex main findings or if a secondary finding is identified are also needed. In addition to being offered as part of medical school and nursing training, education for non-genetic health professionals about genome sequencing technology (eg, basic genetics, processes, limitations and interpreting genetics reports) may be delivered online as long as it includes interactive multimedia components. Education about supporting consent and communicating results to patients may be better delivered through in-person training, and shadowing in the clinic may also be valuable. An important element of training in the future is that genetic experts need to play

**Table 6** Health professionals' views/recommendations for training for non-genetic clinicians

| Training type | Illustrative quote |
|---|---|
| Medical school | "Yeah. I think in the medical school." (Participant 11, non-genetic clinician) |
| Nursing curriculum | "Well I think it will be part of the curriculum in the future. It will just have to become part of nursing curriculum." (Participant 8, consenter) |
| Online training | "…the actual genetic concepts don't necessarily need to be in person they could be done with online modules" (Participant 6, consenter) |
| Shadowing | "you shadow a consultant or a genetic counsellor or someone before you get signed off to go and consent on your own…So when you do the online training, it gives you the knowledge but then when you're doing it practically, it gives you the ability to be able to speak to the patients, answer their questions as the PC doesn't speak to you, doesn't throw in little scenarios whereas in real life we know it doesn't go that straightforward, so doing it with someone and having somebody there actually works" (Participant 10, consenter) |
| Role playing | "talking to a patient, the counselling aspect of it, the understanding of the way that people deal with bad news and understand risk I think is better done in person. …the patient interacting side of it has to be done through face to face training…" (Participant 6, consenter) |
| Day or half-day training | "Yeah, I mean there are all sorts of courses available. The vast majority of clinicians haven't got time to do it and so they're more likely to come to a half day or a 1 day training course than they are to sit down and do some on line training. [some clinicians] haven't got the time to take a year or two out to do a Masters, they want something much more quick and practical and off the shelf and ready now, that just gets them up to speed so that they know enough knowledge to know which test to offer to whom and when." (Participant 14, genetic clinician)<br>"I guess an open training session would be useful, in terms of something like an FAQ sort of session as to what are the expected questions from families and you know how to [answer] them, that sort of a training would be helpful." (Participant 15, non-genetic clinician) |
| Training should be mandatory/national Standard/certification | "Mandatory is a way of keeping on top of who's actually been trained, when they were trained and you get your refresher so, yeah, you do your basic, you get your certificate." (Participant 10, consenter) |
| Genetic clinicians train non-genetic colleagues/buddying systems and genomics champions | "So what we're trying to set up is a kind of a buddying system where we will try and have clinical genetics working with sort of genomics champion in that speciality…. So that the genomics champion from the speciality who will put themselves forward as a sacrificial lamb, and the genetics consultant or genetics counsellor with experience, will kind of discuss and agree what the steps are with feeding that information back to patients. … So what's happened is our genetics teams have gone to the cancer MDTs, disease specific cancer MDTs and they're trying to help people understand about the difference between somatic mutations and germ line mutations. And whether they are clinically actionable… And I think that is working quite well and it's quite labour intensive for the genetics team, but they are you know, working with particular oncologists who want to learn all about this. It's that kind of partnership between genetics and then somebody who's prepared to be the genomics champion from the speciality." (Participant 9, genetic clinician) |

FAQ, frequently asked questions; MDT, multidisciplinary team.

a key role in training and then supporting non-genetic clinicians in understanding sequence data and interpretation of results returned to them from the laboratories and results disclosure to patients.

Our finding that most participants felt health professionals from non-genetic backgrounds should be able to offer genome sequencing with support from genetics colleagues, perhaps reflects the increasing recognition that genetics/genomics is relevant to all areas of medicine and should therefore be 'mainstreamed'.[13–15] As others have concluded, it is inevitable that genomic sequencing will be used more frequently in mainstream clinical practice,[16] and that input from genetic professionals will be needed for more complex cases, in education and for the cross-discipline collaboration that will be essential for classifying variants and understanding phenotypes.[8 16] As the UK Genomic Medicine Service is established, it is important that research continues into how genome sequencing can best be offered, and how genetic and non-genetic professionals can work together to ensure

patients with rare diseases have access to a Genomic Medicine Service that is effective and equitable.

Approaches to the return of secondary findings from genomic sequencing in both research and clinical settings have varied widely, with some programmes choosing to report from a long list of secondary findings and others opting not to offer any secondary findings at all. Professional guidelines for practice also differ. The American College of Medical Genetics (ACMG) recommends that all laboratories conducting clinical genome or exome sequencing for patients should search for DNA variants that are classified as pathogenic in a minimum list of 59 medically actionably genes, and that these secondary findings should be reported to patients, regardless of the original reason for the sequencing being done.[17] Canadian[18] and European guidelines[19] have taken a more conservative approach, and do not recommend secondary findings are looked for in the clinical context at the present time. In the 100 000 Genomes Project, participants could opt to have the project scientists look for secondary findings

in genes on a more limited list than the ACMG guidelines list (at the time of our interviews no secondary findings had been reported to clinicians or patients). The views of our participants regarding secondary findings suggest two possible ways forward. The first option is simply not to offer secondary findings in clinical practice, at least not until further evidence regarding outcomes is obtained. As some of our participants stated, this would be consistent with how many clinical exomes are already managed today. However, as other participants stated, there are potential advantages to offering secondary findings, for example, this could provide much needed evidence on both the outcomes of variants identified incidentally, provided care is taken to ensure the long-term follow-up required to collect such research data. Thus, the second option is to provide the training and resources to be able to offer secondary findings in clinical practice and to encourage patients to consent to research alongside their clinical care. In the words of Les Biesecker,[20] "[a]s a field, we should take full advantage of all opportunities to study these variants by searching them out, returning them to patients and research participants, and studying their utility for predictive medicine."

Strengths of our study include that participants came from several hospitals and had a range of clinical backgrounds, and that all participants' perspectives were informed by actual personal experience of offering genome sequencing to patients within a healthcare service, although as part of a research study. These healthcare professionals were among the first to offer genome sequencing within the UK NHS, and so they have insights that go beyond attitudes expressed in previous studies which mostly occurred before genome sequencing was available and so participants did not have this real-world experience.[21] In some earlier studies, participants' views were often extremely positive, including regarding secondary findings.[22 23] For example, in a 2013 US survey, 96% of clinical genetic healthcare professionals stated they believed secondary findings should be offered to adult patients.[22] The views expressed by our participants are more aligned with the more cautious views expressed by the genetic professionals, who also had experience of delivering genome sequencing where secondary findings were offered, in a recent UK qualitative interview study focused on secondary findings.[11] These differences may, however, also reflect overarching differences in the approach to reporting secondary findings between health professionals in the UK and the USA, where many of the studies to date have been conducted, as the ACMG has advocated the reporting of a wide range of secondary findings from clinical genome sequencing since 2013.

Weaknesses of our study include the small number of participants, meaning the findings may be potentially less generalisable to the wider population of healthcare professionals, and comparisons could not be made between genetic and non-genetic clinicians. In addition, the healthcare professionals were recruited from only two of the 13 GMCs in the UK, and so the findings may not generalise to other geographic locations. The study was also conducted before most of the healthcare professionals had disclosed results to patients, so questions regarding potential benefits are more hypothetical.

In conclusion, genetic and non-genetic healthcare professionals responsible for delivering clinical genome sequencing in the UK have largely positive views of genome sequencing in terms of improving diagnostic yield, but also have concerns and recommendations about practical aspects of delivery. Additional empirical research evidence will be useful to inform policy and practice, including how genome sequencing compares to exome sequencing and the clinical utility of secondary findings.

**Author affiliations**
[1]North East Thames Regional Genetics Service, Great Ormond Street Hospital For Children NHS Foundation Trust, London, UK
[2]Institute of Health Informatics, University College London, London, United Kingdom
[3]Genetics and Genomic Medicine, UCL Great Ormond Street Institute of Child Health, London, United Kingdom
[4]Society and Ethics Research, Wellcome Genome Campus, Hinxton, United Kingdom
[5]Genomics England, Queen Mary University of London, London, United Kingdom
[6]Faculty of Health and Wellbeing, Sheffield Hallam University, Sheffield, United Kingdom
[7]Unique - Understanding Rare Chromosome and Genetic Disorders, Oxted, UK

**Acknowledgements** We gratefully acknowledge the contribution of Andrew Buckton, clinical scientist, and Dominique Stephenson, patient representative, who both gave feedback on an early draft of the manuscript.

**Contributors** SCS, MH and CL were responsible for data acquisition. SCS and MH were responsible for drafting the manuscript and data analysis. CL contributed to data analysis. CL and LSC conceived and designed the study, and contributed to interpretation of data. BS and CP were members of the advisory team who provided feedback on study design, research materials and findings. All authors contributed to revision of the manuscript and approve the final manuscript.

**Funding** This manuscript presents independent research funded by the National Institute for Health Research (NIHR) under the Research for Patient Benefit funding stream (PB-PG-1014-35016: A study to define patient priorities and preferences when consenting to whole genome sequencing to ensure informed choice). MH and LSC are partially funded by the NIHR Great Ormond Street Hospital Biomedical Research Centre. All research at Great Ormond Street Hospital NHS Foundation Trust and UCL Great Ormond Street Institute of Child Health is made possible by the NIHR Great Ormond Street Hospital Biomedical Research Centre. This work has also been supported by Wellcome grant [206194] delivered to CP via Connecting Science, Wellcome Genome Campus. The research was made possible through access to patients being recruited to the 100,000 Genomes Project, which is managed by Genomics England Limited (a wholly owned company of the Department of Health) and is funded by the NIHR and NHS England.

**Disclaimer** The research funded is independent, and the views expressed are those of the authors and not necessarily those of the NHS, the NIHR or the Department of Health.

**Competing interests** CP has been on a secondment with Genomics England as Clinical Lead for Genetic Counselling since October 2016. The other authors declare no competing interests.

**Patient consent for publication** Not required.

**Ethics approval** Ethical approval for this study was obtained from the West Midlands NHS Research Ethics Committee (15/WM/0258).

**Provenance and peer review** Not commissioned; externally peer reviewed.

**Data availability statement** Data are available upon reasonable request.

**ORCID iD**

Melissa Hill http://orcid.org/0000-0003-3900-1425

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
