## [Reviewer comments · BMJ Open]

ARTICLE DETAILS

TITLE (PROVISIONAL)	Delivering genome sequencing in clinical practice: an interview study with healthcare professionals involved in the 100,000 Genomes Project
AUTHORS	Sanderson, Saskia; Hill, Melissa; Patch, Christine; Searle, Beverly; Lewis, Celine; Chitty, Lyn

VERSION 1 – REVIEW

REVIEWER	Sarah Scollon Baylor College of Medicine, USA
REVIEW RETURNED	14-Apr-2019

GENERAL COMMENTS	The manuscript describes the results from a qualitative interview study exploring the perceptions of health professionals involved in the the 100,000 Genomes project on the incorporation of genomic sequencing into clinical practice. This is a well written manuscript with an appropriate study design. The study also takes advantage of a research cohort that can speak specifically to the incorporation of GENOME sequencing as many previous studies have looked at incorporation of exome or large scale genomic sequencing but not solely genomes. Given this, I think an important and significant revision to this paper would be to focus more on that novel aspect of this study and how these results provide new or additional information. Many of the findings reported are the same findings that have been reported in previous studies focused on exome sequencing (benefits of moving science forward versus whether the community is ready and has the time and/or expertise, concerns about this testing in the hands of non-genetics providers, concerns about the return of secondary findings and training needs for professionals, etc). Without the unique aspects of the patient population (genome study and performed in the UK) the findings themselves would be redundant to what has previously been well described. Therefore, this would be a stronger manuscript with a more significant impact on the literature if the authors could place more of a focus on what is new about these findings in the context of genome sequencing. It may also be helpful if the authors acknowledge these similarities to previous studies (most of which are cited by the authors) and explain why their findings in this study population provide new information that can help guide the medical community as we move from exomes to genomes. Despite having these same concerns when exomes were launched, the medical community has moved to wide scale use of exomes in clinical practice so what do we need to know as we prepare to launch genomes. Furthermore, the authors mention that this is one of the first studies to be done in the UK and although they elude to the cultural difference between countries especially as it relates to
--

	secondary findings, further developing this discussion would also make this manuscript more original. A few other general comments include: -It would be helpful for the reader not familiar with the 100,000 genomes project for the authors to provide a little more information about (1) populations being offered testing-pediatric/adult, academic center or general communities, demographics, etc (2) what kind of results are being returned as a part of the study (3) who is returning and how. Also, a little more description on what the Genomic Medicine Service will look like would be helpful. -At line 50 in the data analysis section, the authors do state that the analyses mostly focused on the data around opinions on moving genomic sequencing into clinical practice. It would be helpful to make this more front and center in the analysis section and describe that this is what this manuscript focused on since there is not much data specific to recommendations for the informed consent process itself or return of results. Where it sits now, it is easy to pass this statement over and then the reader is left waiting for the other data. -The authors change between "delivery of genomic sequencing" and "moving genomic sequencing into clinical practice" throughout the paper. The later description seems preferable as the use of "delivery" could be confused at times with delivering the results themselves. Changing this is not an essential adjustment if the authors have purpose behind writing it this way but it is a suggestion that might make the intent of the research question more clear. -Include a more detailed/explicit discussion about the differences between exome and genome sequencing including the pros and cons of each and how these differences impact the consent and return of results process. This can create the groundwork for describing the results as they are specific to genome testing. -Was there any data/quotes where providers with experience with both methodologies addressed these differences or concerns specific to the transition from exome to genome? -Were there any notable differences in the opinions of genetic vs non-genetic providers? -Add a little more information about the difference between the UK and US on secondary finding delivery. Since commercial labs in the US offer all families the option to receive or not receive SFs according to the ACMG recommendations and UK is not returning, this is an important component of the discussion. In the 100,000 Genomes study, were secondary findings returned? -In the discussion the authors expressed that one of the worries reported was clinical departments having adequate time and expertise to analyze and interpret the sequence data. A little more clarification on "clinical departments" and what is actually being interpreted would be helpful-is this referring to raw data? Variant interpretation to decide what goes into a report? A clinical provider trying to make sense of what is on the report? -For the tables with quotes, if a quote is in the written manuscript, it seems redundant to repeat in the table.
--	--

	Overall, this is an important study and, with some reframing of the data in the context of what makes this research different from previous studies, the manuscript would make an important contribution to the literature. Taking the opportunity to compare these results in the context of previous studies could lead to a rich discussion section.
--	---

REVIEWER	Amanda Bergner Columbia University
REVIEW RETURNED	20-Apr-2019

GENERAL COMMENTS	The authors address a timely and important set of questions regarding the transition of genome sequencing from research to clinical settings. The manuscript is well-structured and well-written, and will assist in providing foundational literature on this topic. The inclusion of both genetic and non-genetics clinicians, as well as consenters, helps to broaden the dialogue which will be key as genomic sequencing begins to be offered clinically by non-genetic providers.
---

REVIEWER	Mary Carter University of Bath, UK
REVIEW RETURNED	28-May-2019

GENERAL COMMENTS	This is an interesting paper on a very important topic. I have a few comments and suggestions: RESEARCH ETHICS: More details of the consenting process for interviewees and about the information sent by email to potential interviewees would have been helpful either in the main text or in supplementary material. METHODS: Sampling and recruitment: The second paragraph of the Sampling and recruitment section, which begins 'Participants were purposefully sampled across several different London hospitals to ensure inclusion of all locations' - 'all locations' requires further explanation. The authors do not adequately describe the criteria for stopping recruitment. Patient/public involvement: Despite mentioning that one lay person and three patient support group representatives were involved in the study, only one specific task was mentioned (co-author). A patient perspective may have been helpful in the development of the interview guides or in the analysis of the data. RESULTS: Table 1 Participant Characteristics contains some interesting details about the participants, but could have included more, such as the numbers from each clinical specialty and the various backgrounds of 'consenters' alluded to in the first paragraph of the Results section. An average duration and range for the interviews would also have been helpful information. Numbers of face-to-face and telephone interviews should also have been included. STUDY LIMITATIONS The study limitations described by the authors in the penultimate paragraph of the Discussion section are not mentioned in the
--

	summary (STRENGTHS AND LIMITATIONS OF THIS STUDY) immediately following the Abstract. GENERAL: In some instances, the information presented may have been more logically located elsewhere in the article – for example the last sentence in the Interviews section (about the duration of the interviews) should be in the Results section. The quality of the quotes included in Tables 2-6 is variable – some add insight to the narrative and include the requisite contextual information, while others simply repeat the text provided by the authors.
--	---

VERSION 1 – AUTHOR RESPONSE

Reviewer: 1

The manuscript describes the results from a qualitative interview study exploring the perceptions of health professionals involved in the the 100,000 Genomes project on the incorporation of genomic sequencing into clinical practice. This is a well written manuscript with an appropriate study design. The study also takes advantage of a research cohort that can speak specifically to the incorporation of GENOME sequencing as many previous studies have looked at incorporation of exome or large scale genomic sequencing but not solely genomes. Given this, I think an important and significant revision to this paper would be to focus more on that novel aspect of this study and how these results provide new or additional information. Many of the findings reported are the same findings that have been reported in previous studies focused on exome sequencing (benefits of moving science forward versus whether the community is ready and has the time and/or expertise, concerns about this testing in the hands of non-genetics providers, concerns about the return of secondary findings and training needs for professionals, etc). Without the unique aspects of the patient population (genome study and performed in the UK) the findings themselves would be redundant to what has previously been well described. Therefore, this would be a stronger manuscript with a more significant impact on the literature if the authors could place more of a focus on what is new about these findings in the context of genome sequencing. It may also be helpful if the authors acknowledge these similarities to previous studies (most of which are cited by the authors) and explain why their findings in this study population provide new information that can help guide the medical community as we move from exomes to genomes. Despite having these same concerns when exomes were launched, the medical community has moved to wide scale use of exomes in clinical practice so what do we need to know as we prepare to launch genomes. Furthermore, the authors mention that this is one of the first studies to be done in the UK and although they elude to the cultural difference between countries especially as it relates to secondary findings, further developing this discussion would also make this manuscript more original.

Thank you for this feedback. We have revised the introduction and discussion with the aim of incorporating these suggestions as detailed below.

A few other general comments include:

-It would be helpful for the reader not familiar with the 100,000 genomes project for the authors to provide a little more information about (1) populations being offered testing-pediatric/adult, academic center or general communities, demographics, etc (2) what kind of results are being returned as a part of the study (3) who is returning and how. Also, a little more description on what the Genomic Medicine Service will look like would be helpful.

The introduction has been revised as suggested and now contains additional information on the 100,000 Genomes Project and the NHS Genomic Medicine Service as follows:

“In the National Health Service (NHS) in England a new Genomic Medicine Service was launched in October 2018 that will include offering genome sequencing for patients with undiagnosed rare genetic diseases or a current diagnosis of cancer. A National Test Directory has been developed that describes the full range of tests available. The eligibility, testing and consent materials will be standardised across the Genomic Medicine Service.² The springboard for this new service has been the 100,000 Genomes Project which, under research consent and governance, recruited patients with rare diseases (children and adults) and their relatives, and patients with cancer, for genome sequencing between 2015 and 2018. Recruitment was conducted through thirteen NHS Genomic Medicine Centres located across the UK. In addition to receiving the main findings relevant to the indication for testing, participants could also opt in to secondary findings being looked for by the project scientists. The list of looked for secondary findings includes variants in genes that increase predisposition to breast and ovarian cancer (including BRCA1 and BRCA2), Lynch syndrome (colorectal cancer and other cancers) and familial hypercholesterolemia. Participants could also opt in to secondary analysis of their genomic data to determine carrier status for cystic fibrosis. The return of main and secondary findings from the 100,000 Genomes Project is ongoing and participants are being given their results through NHS clinical pathways.”

-At line 50 in the data analysis section, the authors do state that the analyses mostly focused on the data around opinions on moving genomic sequencing into clinical practice. It would be helpful to make this more front and center in the analysis section and describe that this is what this manuscript focused on since there is not much data specific to recommendations for the informed consent process itself or return of results. Where it sits now, it is easy to pass this statement over and then the reader is left waiting for the other data.

As suggested, we have moved this sentence to the start of the second paragraph of the analysis section.

-The authors change between "delivery of genomic sequencing" and "moving genomic sequencing into clinical practice" throughout the paper. The later description seems preferable as the use of "delivery" could be confused at times with delivering the results themselves. Changing this is not an essential adjustment if the authors have purpose behind writing it this way but it is a suggestion that might make the intent of the research question more clear.

As suggested we have changed “delivery of genomic sequencing” to “moving genomic sequencing into clinical practice” and have changed the term “delivery” through-out when there was potential for confusion for delivery of results.

-Include a more detailed/explicit discussion about the differences between exome and genome sequencing including the pros and cons of each and how these differences impact the consent and return of results process. This can create the groundwork for describing the results as they are specific to genome testing.

A more detailed description of WGS and WES has now been included in the introduction.

“Genome sequencing is increasingly being incorporated into clinical care for the purposes of diagnosing rare disease and some cancers. Genome sequencing allows us to determine most of a person’s complete set of DNA by interrogating both the coding and non-coding regions of the genome. This contrasts with exome sequencing, which only looks at the coding regions of the genome or more targeted approaches that use gene panels to focus on specific sub-sets of candidate

genes. Exome sequencing and panel testing have been widely used in both research and clinical settings, but as costs come down greater use of genome sequencing is occurring. Genome sequencing is more comprehensive than exome sequencing, with wider genome coverage and growing evidence of improved diagnostic yield in some contexts.¹ However, the potential for genomic sequencing to identify a wider range of variants as well as secondary findings that are unrelated to the indication for testing is a key consideration for implementation as this adds to the complexity of consent and return of results processes.”

-Was there any data/quotes where providers with experience with both methodologies addressed these differences or concerns specific to the transition from exome to genome?

Yes – in the description of participant “Concerns” we have noted that some participants felt that the superiority of genome sequencing over exome sequencing had not yet been demonstrated. In addition, some participants noted that use of exome sequencing would obviate the need to make decisions around the return of secondary findings. Both of these points have been illustrated with quotes in the manuscript and included in the discussion.

-Were there any notable differences in the opinions of genetic vs non-genetic providers?

We did not observe any notable differences in the opinions of genetic vs non-genetic providers, but the qualitative study design and associated small sample size meant that the study was not set up to detect differences between groups. One of the limitations of our study, which is noted in the discussion, is the small sample size which does not allow for robust between group comparisons. Views on return of secondary findings were the most divisive in our study, but the differing viewpoints were seen in both the genetic and non-genetic clinician groups.

-Add a little more information about the difference between the UK and US on secondary finding delivery. Since commercial labs in the US offer all families the option to receive or not receive SFs according to the ACMG recommendations and UK is not returning, this is an important component of the discussion. In the 100,000 Genomes study, were secondary findings returned?

Participants in the 100,000 Genomes Project could opt in to receive a limited number of secondary findings, far fewer than the list of 59 specified in the ACMG guidelines. More detailed information on which secondary findings were offered as part of the 100,000 genomes project has been added to the introduction (see above). In the discussion we have added more detail on the wide variation in approaches used for return of secondary findings, both within and between countries.

“Approaches to the return of secondary findings from genomic sequencing in both research and clinical settings have varied widely, with some programmes choosing to report from a long list of secondary findings, and others opting not to offer any secondary findings at all. Professional guidelines for practice also differ. The American College of Medical Genetics (ACMG) recommends that all laboratories conducting clinical genome or exome sequencing for patients should search for DNA variants that are classified as pathogenic in a minimum list of 59 medically actionably genes, and that these secondary findings should be reported to patients, regardless of the original reason for the sequencing being done.¹⁷ Canadian¹⁸ and European guidelines¹⁹ have taken a more conservative approach, and do not recommend secondary findings are looked for in the clinical context at the present time. In the 100,000 Genomes Project, participants could opt to have the project scientists look for secondary findings in genes on a more limited list than the ACMG guidelines list (at the time of our interviews no secondary findings had been reported to clinicians or patients).”

-In the discussion the authors expressed that one of the worries reported was clinical departments having adequate time and expertise to analyze and interpret the sequence data. A little more

clarification on "clinical departments" and what is actually being interpreted would be helpful-is this referring to raw data? Variant interpretation to decide what goes into a report? A clinical provider trying to make sense of what is on the report?

This sentence has been revised to clarify the specific concerns participants raised about resource demands on clinical departments as follows:

“Some participants’ worries were primarily about whether healthcare professionals in genetics and non-genetics departments have adequate resources to explore the tiered variant report that is returned, undertake variant interpretation, conduct additional phenotyping and technically validate results to issue a report.”

-For the tables with quotes, if a quote is in the written manuscript, it seems redundant to repeat in the table.

Repeated quotes have been removed from the table.

Overall, this is an important study and, with some reframing of the data in the context of what makes this research different from previous studies, the manuscript would make an important contribution to the literature. Taking the opportunity to compare these results in the context of previous studies could lead to a rich discussion section.

Thank you for your positive feedback and suggestions to improve the manuscript.

Reviewer: 2

The authors address a timely and important set of questions regarding the transition of genome sequencing from research to clinical settings. The manuscript is well-structured and well-written, and will assist in providing foundational literature on this topic. The inclusion of both genetic and non-genetics clinicians, as well as consenters, helps to broaden the dialogue which will be key as genomic sequencing begins to be offered clinically by non-genetic providers.

Thank you for your positive comments about our manuscript.

Reviewer: 3

This is an interesting paper on a very important topic. I have a few comments and suggestions:

RESEARCH ETHICS:

More details of the consenting process for interviewees and about the information sent by email to potential interviewees would have been helpful either in the main text or in supplementary material.

A sentence has been added to the section titled “sampling and recruitment” to note that written informed consent was obtained prior to commencing the interview as follows:

“Written consent was obtained prior to the interviews to ensure informed and voluntary participation.”

We have also now included the information sent to potential participants when inviting them to participate in the supplementary material.

METHODS:

Sampling and recruitment:

The second paragraph of the Sampling and recruitment section, which begins 'Participants were purposefully sampled across several different London hospitals to ensure inclusion of all locations' - 'all locations' requires further explanation.

This sentence has been revised as follows:

"Participants were purposefully sampled across four London hospitals to ensure inclusion of participants from different geographical locations, various approaches to recruitment, and clinicians from a range of clinical backgrounds."

The authors do not adequately describe the criteria for stopping recruitment.

A sentence has been added to the section titled "sampling and recruitment" to describe the decision to stop recruitment as follows:

"Interviews and analysis were undertaken concurrently, and recruitment ceased when no new themes were emerging during the interviews."

Patient/public involvement:

Despite mentioning that one lay person and three patient support group representatives were involved in the study, only one specific task was mentioned (co-author). A patient perspective may have been helpful in the development of the interview guides or in the analysis of the data.

We have added a sentence to the PPI section to explain the role of the advisory team as follows:

"The advisory team for this study had one lay-person and three patient support group representatives; including co-author BS, who has also contributed to the revision of the manuscript. Over the course of the study the advisory team has provided ongoing review and feedback on study conduct, research materials, interpretation of data and reporting of findings."

RESULTS:

Table 1 Participant Characteristics contains some interesting details about the participants, but could have included more, such as the numbers from each clinical specialty and the various backgrounds of 'consenters' alluded to in the first paragraph of the Results section. An average duration and range for the interviews would also have been helpful information. Numbers of face-to-face and telephone interviews should also have been included.

As suggested, we have added the numbers of participants from each clinical specialties and consentor background to Table 1. The number of face to face and phone interviews and the range and median interview length has been added to the text of the results section as follows:

"There were seven telephone and 16 face-to-face interviews. The interviews ranged in length from 19 to 75 minutes (median = 34 minutes)."

STUDY LIMITATIONS

The study limitations described by the authors in the penultimate paragraph of the Discussion section are not mentioned in the summary (STRENGTHS AND LIMITATIONS OF THIS STUDY) immediately following the Abstract.

Two points in the summary have been revised so that the summary now covers all of the study limitations set out in the Discussion.

- The qualitative methodology employed in this study facilitated an in-depth exploration of interviewees' perspectives and allowed them to express their views in their own words; however, the sample size was small and encompassed only four London hospitals, which brings limitations for wider generalisability and did not allow between group comparisons
- Interviewees' views were informed by their experience of actually offering genome sequencing to NHS patients, rather than views being entirely based on hypothetical scenarios; however, at the time of the interviews very few results had been returned from the 100,000 Genomes Project

GENERAL:

In some instances, the information presented may have been more logically located elsewhere in the article – for example the last sentence in the Interviews section (about the duration of the interviews) should be in the Results section. The quality of the quotes included in Tables 2-6 is variable – some add insight to the narrative and include the requisite contextual information, while others simply repeat the text provided by the authors.

As suggested we have added a description of the interview duration to the Results section.

We have reviewed the quotes included in the tables and in those cases where a quote simply repeated the main text we have either selected alternative quotes or expanded the quote to add insight and context.

VERSION 2 – REVIEW

REVIEWER	Sarah Scollon Baylor College of Medicine, USA
REVIEW RETURNED	30-Jul-2019

GENERAL COMMENTS	The authors have addressed the comments from the reviewer specifically related to more details around the context of the research study in which the genomic sequencing was completed as well as highlighting any differences these results have in the context of a genome sequencing only study versus other studies including exome and genome sequencing. The findings themselves are not that unique compared to previously studies and I would defer to the editor on how much this weighs into the publication decision. However, the important points of this paper include that similar findings were replicated in a UK study looking at genome only testing. Therefore it doesn't appear that the use of genomes vs exomes significantly altered provider opinion. It also highlights some of the different views on secondary findings between various countries.
---

REVIEWER	Mary Carter University of Bath
REVIEW RETURNED	22-Jul-2019

GENERAL COMMENTS	This revised version includes useful additional information about the participants, interviews, patient/public involvement in the study and approach to stopping recruitment. It is a well-written and very readable article concerning an important research topic.
--